# Experimental and Theoretical Study of Plastic Deformation of Epoxy Coatings on Metal Substrates Using the Acoustic Emission Method

**DOI:** 10.3390/ma15113791

**Published:** 2022-05-26

**Authors:** Petr Louda, Aleksandr Sharko, Dmitry Stepanchikov, Artem Sharko

**Affiliations:** 1Department of Material Science, Technical University of Liberec, 461 17 Liberec, Czech Republic; petr.louda@tul.cz; 2Department of Transport Technology and Mechanical Engineering, Kherson Marine Academy, 73000 Kherson, Ukraine; alexandr.vladimirovich.sharko@gmail.com; 3Department of Energetics, Electrical Engineering and Physics, Kherson National Technical University, 73008 Kherson, Ukraine; dmitro_step75@ukr.net

**Keywords:** plastic deformation, acoustic emission, signal processing, principal component analysis

## Abstract

Propagation of acoustic emission signals in continuous conjugated media under real-time loading was explored. The results of explored plastic deformation polymer coatings on a metal base using the acoustic emission method with synchronization of deformations and the moments of occurrence of acoustic emission signals are presented. Using the principal component method, the acoustic emission spectra, which make it possible to trace the evolution of deformation transformation processes, were analyzed. Presented the results of theoretical and experimental studies on the separate propagation of acoustic emission vibrations in a polymer coating, a metal base, and their joint combination in the form of multilayer structures. Boundary problems of propagation of acoustic emission signals in the conjugation of continuous media are considered from the standpoint of an elastic continuum and wave representations. The main variables are the force that initiates the appearance of acoustic emission signals and the displacement that determines the propagation of elastic waves. Based on the local rearrangement of the internal structure of conjugated media under conditions of development of deformation processes in the material, the verification of the main theoretical models of energy spectrum acoustic signals in continuous media at the micro-, meso-, and macro-levels was carried out. In this work, we present experimental data on a set of basic acoustic emission characteristics for four-point bending. It is shown that the principal components method reduces the dimension of data while maintaining the least amount of new information. Using the method of principal components to determine the stages of plastic deformation of polymer coatings on a metal base using the acoustic emission method. With the digitalization of acoustic emission signals and noise filtering, new possibilities for isolating a weak signal at the noise level appear even when its amplitude is significantly lower than the noise level. The study results can be used to predict the degree of destruction of two-layer materials under loading.

## 1. Introduction

Polymer coatings are used to protect equipment from corrosion damage and destruction. During the operation of protective coatings on a metal base, it becomes necessary to control the state of both the metal base and the coating since damage to the protective layer leads to metal corrosion. Despite a wide range of polymer composite coatings, the most common are anti-corrosion adhesives based on epoxy oligomers. The acoustic emission (AE) method makes it possible to single out the stages of plastic deformation of materials, each of which is a scale level with its deformation mechanism. To obtain an appropriate information base, experimental studies on the deformation of materials with simultaneous recording of acoustic signals in real-time are required.

The relationship between the processes occurring in the structure of materials under loading and the accompanying AE effects makes it possible to predict changes in materials’ mechanical properties and structure based on acoustic measurements [1,2]. A literature review is presented, with a general target orientation and practical implementation on experimental and theoretical studies of plastic deformation of polymer coatings on a metal basis using the acoustic emission method.

The problem diagnostics and identification form of the structure of materials are solved by studying the effects and mechanisms of generation and propagation of AE signals that change their parameters under loading. In [3], the behavior of AE signals in aluminum alloys during cyclic loading is presented in [4] during the bending of steel sheets with the distribution of residual stresses. In [5,6], the results of recognition of changes in the structure materials during destruction according to acoustic emission data are presented. In [7,8,9], an information-entropy model of the quality of input information is presented, and in [10] the most informative parameters of AE signals under conditions of bending deformation and uniaxial loading are established.

The study of the kinetics of deformation materials under load is based on the analysis of response to an external force.

In [11,12], the results of experimental studies on the identification of the propagation of acoustic emission oscillations in an epoxy polymer matrix are presented. These results will be used to apply this material as a coating. In [13], the results of identification of the damage mechanism in composites using the AE phenomenon are presented. Identification of the deformation stages, initial cracks, and fatigue damages is carried out in [14,15,16,17].

It should be noted that only homogeneous materials without coatings were studied. Studies of multilayer structures using the AE method have not been carried out before. This is the novelty of the review.

The technology for determining the residual resource of metal products under combined load conditions by AE measurements is presented in [18]. In [19,20], criteria for predicting the destruction of a polymer composite material based on the total accumulation of AE acts, the probability density of peak amplitudes, and median frequencies of AE signals based on wavelet decomposition were proposed.

In [21], the results of studies of plastic deformation and the stress–strain state of an aluminum alloy using wavelet transforms of AE signals are presented. The effectiveness of the proposed method lies in the fact that with the help of the wavelet transform mechanism, it becomes possible to establish a signal correlation at the input of the receiving transducer with a remote source. The information contained in AE signals can be represented by a whole set of characteristics of amplitude, time, energy, and frequency parameters and their manifestations in the form of average and maximum amplitude, amplitude dispersion, intervals between pulses, signal observation time, signal energy and density, maximum amplitude in frequency area, frequency localization of the main AE peak.

To reduce the data dimension, losing the least amount of information allows for the principal components method. When analyzing AE signals, the features of AE spectra are often highly dependent on each other, and their simultaneous presence is redundant. The method is based on the idea of constructing a linear feature transformation, in which the signal-to-noise ratio in the new coordinate system decreases with increasing coordinate numbers. The data are presented as the sum of the useful signal and the uncorrelated noise in this case. The principal component analysis is designed to extract uncorrelated feature combinations from correlated data. As follows from the above review of practical applications of the AE method, the novelty and relevance of this work lie in the study of deformations on coated materials and the use of effective methods for processing AE information.

In [22], the identification of AE bursts coming from triaxial tests during the deformation of steel pipes with a protective coating is presented by classifying the physico-statistical parameters obtained by the principal component method. In [23], the results of applying the principal component method to the study of AE signals when the stress–strain state of aluminum alloys changes are described. Digitalization results are a necessary stage of mathematical processing of information support processes of plastic deformation and destruction of materials in real-time. In [24], the results of applying the principal component method to the study of AE during plastic deformation of lead alloys are described. For a quantitative analysis of the influence of the stages of strain hardening on acoustic emission, clustering of experimental AE signals was used. 

In [25,26], the principal component method was used to reduce the size of AE data when drilling composite materials. In [27], the AE wave clustering method was used to monitor the state of structures during bending tests, in [28] for compression, in [29,30] to monitor the development of damage during static and dynamic tests, in [31] for torsion tests three-dimensional braided composite rolls. The AE signals were analyzed by the methods of principal components and fuzzy clustering.

Cluster analysis of AE signals in combination with infrared thermography was used in [32] to analyze the evolution of defects in glass–epoxy composites. When analyzing AE signals, the k-means method and the principal component method were used. It is shown that the AE signals during tensile loading can be divided into four clusters of damage types, such as matrix cracking, fiber detachment from the substrate, delamination, and fiber detachment. In [33], clustering of features of damage mechanisms was used in the processing of AE data in the time and frequency domains. In [34], the technology of neural networks with a self-organizing display of signs of the distribution of the characteristic parameters of AE was used to assess corrosion damage on prestressed steel filaments. Cluster analysis of AE signals made it possible to identify four types of damage sources and evaluate their evolution.

The principal component method makes it possible to visualize data, simplify calculations and interpret compressed volumes of input information. The proposed approach can be applied in the study of plastic deformation processes to diagnose the deformation behavior of materials. With regard to the analysis of the features of the study of AE signals under loading of multilayer structures, it will allow us to discover new effects that occur when they are loaded. Among all the analyzed works, our studies are new and unparalleled.

The solution to the scientific problem of diagnostics and identification of stages of change in the structural states of epoxy coatings on a steel substrate under loading consists not only in the use of new samples of measuring equipment but also in the use of new forms of processing informative parameters that are sensitive to changes in the structure of materials through the use of the method of the principal component.

The scientific direction of the proposed work lies in a more profound use of the principal component method in assessing the stages of deformations of multilayer structures and determining the boundaries of their changes.

This work aims to study the structural features and characteristics during the deformation of epoxy coatings on steel substrate in developing an analytical apparatus for identifying the state of materials.

## 2. Materials Methods 

Samples of St3sp steel used as a base and ED-20 epoxy resin used as a coating were used as research materials. The choice of epoxy resin as the research material is explained by the fact that it is characterized by high adhesion to metal, glass, and concrete [35]. One of the common structural carbon steels St3sp was chosen as the material used for the substrate. Mechanical properties and international analogs are presented in [1,2].

Acoustic emission measurements of the stress–strain state of continuous media under real-time loading were used as research methods [36,37]. The method of principal components was used to process the results of acoustic measurements.

## 3. Theoretical Model

Modern mathematical models of the energy spectrum of AE signals take into account the presence of several scales in the medium, their coordinated interaction, and the possibility of energy transfer from one level to another at the micro-, meso-, and macro-structural levels. Interest in solving the problem of studying the plastic deformation of polymer coatings on a metal base using the AE method requires a more detailed study of the dependence of the properties of materials on their microstructure. Suppose deformation processes in materials can be described on the basis of loading diagrams. In that case, the evolution of the defective structure of metallic materials during the formation of AE signals can only be described at the atomic level [38,39].

The discrepancy between the coefficients of thermal expansion coating and base materials leads to the appearance stresses and their interfaces, which causes damage to the coating, the formation of cracks, and other defects. Finding the energy spectrum of AE signals in continuous conjugated media is a means of predicting the destruction of materials [40,41,42].

Operator beams in studying the spectral properties of AE signals in the problems of motion of structural imperfections subjected to bending forces are considered in [35].

The model of a two-layer conjugated medium in the form of a linear chain of atoms with a simple single point defect is shown in Figure 1.

When pairing two continuous media in each of them, it is possible to single out the boundary regions S and S*, in which acoustic signals propagate. They have dimensions depending on the mechanical parameters of the medium. Further, the signals do not propagate due to attenuation.

The equation of motion of the defective zone has the form:(1)$$\left\{\begin{array}{l} \frac{\partial \sigma_{x}}{\partial x}+\frac{\partial \tau_{xy}}{\partial y}+\frac{\partial \tau_{xz}}{\partial z}+\rho f_{x}=\rho \frac{d\upsilon_{x}}{dt},\\ \frac{\partial \tau_{xy}}{\partial x}+\frac{\partial \sigma_{y}}{\partial y}+\frac{\partial \tau_{yz}}{\partial z}+\rho f_{y}=\rho \frac{d\upsilon_{y}}{dt},\\ \frac{\partial \tau_{xz}}{\partial x}+\frac{\partial \tau_{yz}}{\partial y}+\frac{\partial \sigma_{z}}{\partial z}+\rho f_{z}=\rho \frac{d\upsilon_{z}}{dt}. \end{array}\right.$$
where *ρ*-defect zone material density, *υ*-the speed of its particles, *f*–density of mass forces acting on an elementary volume *V*, *σ_x_*, *σ_y_*, *σ_z_*—normal stress tensor components, *τ_x_*, *τ_y_*, *τ_z_*—shear stress tensor components.

By the local properties of functions, one can judge the global properties of the object’s materials. The properties of beams are determined by the properties of tensor fields of local manifolds [43]. Under diagnostic conditions, can be transferred to the properties of deeper global structural changes in materials during bending of continuous conjugated media. A useful tool for such transformations is the theory of sheaves. The theory of sheaves is a special mathematical apparatus that provides a unified approach for establishing a connection between local and global properties of the topology of spaces. Sheaves play a significant role in the topology of differential and algebraic geometry and are used in number theory and category analysis. A remarkable property of the theory of beams and a developed mathematical apparatus can serve as a basis for their further possible use in analyzing AE signals during the bending of multilayer structural materials. The beam on the topological space of multilayer products can find its development in the formation of defects in the experimental and theoretical study of the plastic deformation of epoxy coatings on metal substrates using the acoustic emission method.

The equations of motion of particles in the regions S and S* (Figure 1) have the form
(2)$$-\omega^{2}\rho_{S}u_{S}+S\Phi u=q_{S}$$
(3)$$-\omega^{2}\rho_{S*}u_{S*}+S*\Phi u=q_{S*}$$
where *ρ_S_* and *ρ_S*_* material density of both media, *q_s_* and *q_s*_*—interaction forces, Φ—elastic energy operator

For a defect-free structure of the material, when the elastic bonds between the atoms of both media are stable, we have a static problem that can be reduced to an integral Fredholm equation of the second kind:(4)$$y(x)=\lambda \int_{a}^{b}k(x,t)y(t)dt+f(x)$$
where *k*(*x*, *t*)—integral equation kernel, *λ*—characteristic number, *a*, *b*—integration limits, *f*(*x*)—free member.

The area of application of the elastic energy operator is determined by the propagation zone of acoustic signals in both media *S* and *S*^*^, limited on the one hand by the damping of oscillations, on the other hand by the mechanical properties of the structure of materials *H* and *H*^*^, which are determined by the boundary conditions for the existence of signals *a*_t_ ≤ t ≤ *b*_t_ and geometric dimensions of the medium *a_x_* ≤ *x* ≤ *b_x_* (Figure 2).

Aggregate *λ* determines the energy spectrum of the elastic energy operator. Value *y*(*t*) characterizes the receiving signal, and *y*(*x*)-triggered signal.

The solution of the boundary value problem of the structure of the energy spectrum of the AE signal is given by the formula
(5)$$y\left(x\right)=\int_{a}^{b}G\left(x,\omega \right)f\left(\omega \right)d\omega$$
where *G*(*x*,*ω*)–Green’s function.

The rationale for using Green’s function to determine the energy spectrum of AE signals in continuous conjugated media lies in its physical meaning-this is a displacement field created by a single force that causes the formation of structural defects and AE signals acting on one atom of the cell.

Green’s function is the kernel of the operator inverse to the elastic energy operator Φ. The construction of Green’s function in the theory of defect-free structures was carried out in [41]. With some assumptions set out in [42], this formalism can be transferred to the definition of the energy spectrum of AE signals in continuous conjugated media.
(6)$$G\left(x,\omega \right)=\frac{1}{2\pi }\int \frac{e^{ikx}}{\Phi \left(k,\omega \right)}dk=\frac{1}{2\pi \rho }\int \frac{e^{ikx}}{\omega^{2}\left(k\right)-\omega^{2}}dk$$

To obtain a solution to the integral equation of motion of particles in conjugate media, we reduce it to a boundary value problem of a linear operator using the Fourier transform.
(7)$$f\left(y\right)=\frac{1}{\sqrt{2\pi }}\int_{-\infty }^{+\infty }e^{-i\omega t}f\left(\omega \right)d\omega$$

Each linear operator corresponds to a linear equation, on the basis of which its eigenvalues are found.

Thus, the methodological basis for determining the energy spectrum of AE signals in conjugated media is the use of Green’s function and the Fourier transform. It follows that the better the function is concentrated in time, the more it is blurred in frequency. With an increase in the applied voltage, high-amplitude acoustic signals are formed, which correlates with deformation jumps.

The value λ included in the Fredholm equation denotes the variety of energy spectrum components that can be recorded by various AE measurement methods.

The initialization of AE signals under the loading of a material is considered using the evolutionary concepts of the theory of dislocations. The presented solution of the equation of motion of a system of conjugate media during the development of internal structural defects allows us to perform a spectral analysis of differential operators of elastic energy and their distribution over eigenvalues.

## 4. Experiment

For polymer coatings based on a metal-based epoxy matrix, the situation is complicated by generating AE signals from materials of different physical natures. In addition, the application of the AE phenomenon, which accompanies the processes of various types of loading and control of technological processes associated with a change in the structure of materials, reveals great technical difficulties associated with the variety of recommended information parameters and their representations for further processing and use.

The test specimens were cut from sheet metal 200 × 20 × 3 mm in size. ED-20 resin with polyethylenepolyamine (PEPA) hardener was used as a protective coating applied on one side of the sample. The coating thickness was 1 mm.

In the experiment, the coating thickness was fixed. As follows from numerous literature sources, there is a relationship between the plasticity of the metal substrate and damage to the coating under loading, due to the thickness of the coating. The consequence of this is cracking on the coating, interfacial delamination, microcracks, etc. This is an important scientific area of research in the field of materials science, structural mechanics, and the theory of strength of conjugated media. Its consideration goes beyond the scope of more than one similar scientific article and will undoubtedly be the basis for further research.

Epoxy resin ED-20 was used to form the epoxy matrix. For crosslinking the epoxy binder, cold curing hardener PEPA was used, the content of which was q = 10 mass.h. (indicated per 100 mass parts of epoxy resin ED-20).

The formation of the epoxy matrix was used in the following sequence: heating the resin to a temperature T = 353 ± 2 K and holding at a given temperature for a period of time τ = 20 ± 0.1 min; ultrasonic treatment of the composition over time τ = 1.5 ± 0.1 min; cooling the composition to room temperature over time τ = 60 ± 5 min; and the introduction of the hardener and mixing the composition over time τ = 5 ± 0.1 min.

The materials were cured according to the experimentally established regime: the formation of samples and their exposure for an hour *t* = 12.0 ± 0.1 h at a temperature T = 293 ± 2 K, heating at a rate *v* = 3 K/min up to temperature T = 393 ± 2 K, exposure of samples at a given temperature for a period of time *t* = 2.0 ± 0.05 h, free cooling to temperature T = 293 ± 2 K for a time *t* = 24 h on air.

Four-point bending tests were used to evaluate bending stresses.

The experimental setup used for measurements is shown in Figure 3. The block diagram of the installation includes: 1-force-measuring mechanism, 2-deformation mechanism, 3-controlled sample, 4-fixed prism, 5-indenter, 6-filter, 7-analog-to-digital converter, 8-information accumulation and conversion unit, 9-recording device, 10-piezoelectric sensor, 11-strain gauge, 12-pre-amplification unit (Figure 3a).

The functional diagram of the experimental setup is shown in Figure 3b. The sample (1) is installed between the prisms of the testing machine (2) and an indenter in the form of a loading device rod (3). The loading device (4) consists of a pulsed DC power supply unit (5), a drive unit (6), and a planetary gearbox (7). A counter for counting the time from the start of loading (8) is installed on the drive unit (6). An acoustic emission sensor (12) is installed directly on the sample. To assess the degree of loading of the sample, simultaneously with the timer (8), the signal is fed to the strain gauge (9) and to the digital deflection indicator (10), and then through the microprocessor (11) to the computer (16). A signal from the AE sensor (12) is also fed here through the information unit (13), the unit for accumulating and processing information (14), and the AE recording device (15). The recording device can be made in the form of a storage oscilloscope. Thus, the computer collects simultaneously information from the parameters of the loading device, deflection, and the moments of occurrence of the AE signals as a function of time.

The measurement process was carried out in combination with mechanical tests and acoustic emission measurements. At the same time, the process of registering loading, deflection and time changes, as well as fixing AE signals, was fully automated by means of synchronous processing of the received data from the moment the machine drive was activated. A general view of the setup for AE measurements for four-point bending is shown in Figure 3c.

Measurement of the deformation of the samples under uniaxial loading was determined by fixing the elongation using a Micron digital indicator DT-7011 micrometric electronic displacement indicator. The micrometric electronic indicator of displacement, in contrast to the mechanical one, is adjusted to zero at any point of the scale, so that you can observe the deviation from the dimensions of the test sample in the range of 0–12.7 mm with an inaccuracy of 0.001 mm. According to [36], bending tests are carried out on machines operating on the principle of specified deformation. Recommended moving speed of the indenter is (0.5 ± 0.1) mm/min.

For acoustic measurements, broadband sensors to the AE of the AF-15 device with a bandwidth of 0.2 … 0.5 and 0.2 … 2.0 MHz were used. The information-measuring system used in the experiment provided the indication and registration of AE signals with their subsequent storage in the computer memory using a RIGOL DS1052E digital oscilloscope.

To improve the accuracy of measurements during dynamic bending tests, the installation is supplemented with a loading synchronization system with the registration of the AE signal coupled with a personal computer. The computer simultaneously receives information from the parameters of the loading device, deflection, and moments of the occurrence of AE signals as a function of time. During the tests, information about the deformation of the sample and the amplitude of the AE signals was recorded in a file with a given periodicity. The tests were carried out with registration of the load and the corresponding deformation with simultaneous fixation of the moments of occurrence of AE signals in compliance with standards [37]. AE sensors were installed both on the side of the metal base and on the side of the composite coating.

To clarify the question that the recorded AE signals refer specifically to the coated samples, and not to the metal substrate, it is necessary to compare the results of measurements performed separately on the metal substrate, on the epoxy coating, and on their conjugation under identical measurement conditions. The results of AE measurements synchronized with mechanical tests at four-point bending, performed for a polymer matrix used in the form of coatings, are shown in Figure 4.

Similar complex AE measurements and mechanical tests performed for the metal base of ST3sp steel are shown in Figure 5.

Visualization of the measurement results of coated samples at different loads is shown in Figure 6.

The results of comparative AE measurements and mechanical tests shown in Figure 4, Figure 5 and Figure 6 were performed while maintaining the constancy of experimental conditions on the setup shown in Figure 3 with the location of AE sensors between the prisms of the indenter’s loading device on the lower side of the substrate. The correlation between the theoretical model and the experiment carried out on coated samples manifests itself in the establishment of deformation jumps and in the formation of high-amplitude AE pulses with increasing load within the loading diagram, as well as in establishing a correspondence between the characteristics of the discrete structure of materials in contact media and the propagation parameters of AE signals.

## 5. Results and Discussion

The location of the signals corresponds to the order in which they occur under loading. The peculiarity and originality of such a presentation of AE signals in Figure 6 lie in the desire to display not so much the dynamics of the sequence of the loading process by the force-measuring device, but the synchronous display of the moments of visualization of AE bursts by the deflection indicator, and the priority in AE measurements were given to direct tracking of changes in plastic deformation.

Figure 7 shows on one scale the entire set of AE signals obtained in the order of their occurrence and fixation during sample loading, which gives a visual representation of the change in peak amplitudes. Table 1 presents the experimental values of the main characteristics of the four-point bending of the test sample.

Relationship between the processes of defect formation and the occurrence of AE signals makes it possible to determine the degree of change in the structure of materials under load. The most difficult moment in studying the behavior of materials under load is the identification of AE signals that characterize the state and mechanical properties of materials directly at the source of initiation of AE signals. Information about the source of AE signals can be obtained not only by measurements but also by integrating theoretical models of AE occurrence.

According to the definition, plastic deformation is an irreversible deformation in which, after the end of the applied forces, an irreversible displacement of interatomic bonds occurs.

The mechanisms of plastic deformation can be considered depending on the scale levels on the basis of mesoconcepts about the dislocation structure of materials [20,21].

The mechanism of plastic flow at the mesoscale level is based on energy dissipation, which leads to discontinuity with the appearance of cracks in the surface layer during the development of mesobands of localized deformation in the metal substrate. Mesoconcentrators of stresses in coated materials arise at their interface, developing towards the surface of the sample due to inconsistent shear deformation of mating media.

The mechanism of plastic deformation stages requires the use of continuum mechanics and dislocation theory approaches. Plastic deformation in continuum mechanics takes into account the translational motion of defects under the action of an applied stress. The plastic flow curve is obtained by calculating the strain hardening above the yield strength of the material. The theory of dislocations describes the microscopic behavior of a deformable body on the basis of the mechanisms of generation of elementary acts of plastic shear and their dislocation ensembles.

The translational nature of plastic deformation as a multilevel system underlies a new scientific direction in strength physics and materials science—the physical mesomechanics of a deformable solid [44,45].

The mechanisms of plastic flow and the corresponding stages of the hardening curve obey the principle of scale invariance. The hierarchy of structural levels of plastic deformation of solids can vary from three stages, including elastic deformation, elastic–plastic and fracture, to five or more, depending on the structure of the material.

For St3sp steel coated with epoxy resin ED-20, plastic deformation at the initial stage occurs during the propagation of the Lueders band front. At the same time, the coating prevents the fronts of the Lueders bands from propagating at a constant speed and the movement occurs abruptly at distances equal to the cracking period. The velocity of the front of the Luders strip is maximum in the initial period of crack formation and is equal to zero at the moment before the formation of the next transverse crack in the coating.

The frontal nature of the Luders band propagation is characterized by the formation of an extended yield plateau. The accumulation of the bending moment during the propagation of the Lueders band in the metal substrate leads to the formation of elastoplastic fracture precursors in the coating. The development and accumulation of damage in contiguous media caused by deformation processes are fixed by a change in the surface structure. According to the state of the surface, it is possible to assess the degree of deformation, moments, places of its localization, and degradation [46,47].

In industry, technological tests are used, the task of which is to evaluate the plasticity of deformed semi-finished products. The criterion for the suitability of products can be a given bending angle and the appearance of the first crack after bending at an angle equal to or greater than the specified one, as well as the possibility of bending the plate to a parallel state.

Flexural tests are aimed at evaluating the transition from brittle to ductile. For a coated metal material, the bending diagram allows you to determine the magnitude of the loads corresponding to the limits of proportionality, elasticity, yield, and bending strength. When testing for bending medium-carbon steels, the diagram σ, ε does not have a pronounced yield point. In this case, the yield strength is determined by a certain tolerance through the elastic modulus of the material.

The resistance to plastic deformation is determined from the deformation diagram in coordinates σ, ε. The stress required to create plastic deformation determines the yield strength. The strain hardening intensity is expressed in terms of the strain hardening coefficient K = dσ/dε.

During plastic deformation of epoxy coatings on metal substrates using the acoustic emission method, a successive change in the stages of plastic deformation associated with the features of plastic flow is observed. To determine the boundaries of the stages of plastic deformation, the values of the strain hardening coefficient were calculated from the measured values of the load and deflection. From the inflections of the dependences of the values of this coefficient on strain, it is possible to more correctly determine the boundaries of the stages than directly from the stress–strain curve [21]. According to the inflections of the dependences of the strain hardening coefficients, four stages of plastic deformation were identified, the boundaries of which are characterized by their values of the strain hardening coefficient.

During the experiments, five identical coated samples were tested. In this case, AE signals arose in certain zones of the loading diagram with a particular spread. Considering this scatter and instrumental errors in measuring mechanical stress and deflection, the total absolute errors in determining mechanical stress and deflection were calculated for a confidence level of 0.98. The maximum standard deviation for mechanical stress ±9.529 Mpa, for deflection ±0.215 mm.

Figure 8 shows the loading diagram of the sample under study made of St3sp steel with an epoxy coating based on ED-20 resin.

Figure 9 shows microscopic images of the cross-section of coated samples for various stages of plastic deformation under loading. Microscopic photographs were taken at an angle so that the structure of the coating layer was visible. Yellow marks indicate the contour of the sample, on which the dimensions of the sample itself are indicated, the thickness of the base is 3 mm, and the thickness of the coating is 1 mm. At the same time, due to the tilt angle of the microscope, the depth is also visible in the photograph (sample width 20 mm).

The measurement results are shown in Figure 8 as intervals for each series of recorded AE signals. The AE signals presented in Figure 6 and Figure 7 and in Table 1 correspond to a sample that gave results close to the middle of the obtained intervals.

Acoustic emission is an energy indicator of the processes occurring during the restructuring of the structure caused by plastic deformation under loading.

A joint analysis of the strain hardening curve, the evolution of the deformation structure under loading, and the established patterns of changes in AE signals can serve as better estimates of the transition from one stage of hardening to another.

The results of constructing the deformation curve according to data of the force-measuring device when changing the deflection Δ*L* as a dependency *K = f(**σ)* are also presented in Figure 8. From the consideration of the given graphs, four successively alternating stages of strain hardening can be distinguished, at each of which energy is accumulated by the structure with the emission of AE pulses due to translational shifts.

Stage I represents an almost linear section of purely elastic deformation with a nonlinear decrease in the coefficient *K*, stage II—the beginning of plastic deformation, the observed decrease in the slope of the curve *σ = f(*Δ*L),* and transition coefficient K to the horizontal zoom area Δ*L*, stage III—continuing decrease in the slope of the curve *σ = f(*Δ*L)* and weak dependence of the coefficient *K* from ΔL, stage IV—pre-fracture and a sharp decrease in the coefficient of strain hardening, at which the coefficient *K* changes sign.

The nature of the stages of plastic deformation is explained by the internal structure of the material. The appearance of stage I is due to the onset of the formation of dislocation structures. This stage is associated with uniform plastic deformation. Stage II is characterized by the formation of individual spatially distributed mesobands along the surface of plastic deformation, associated with the emergence of deformations at the interfaces between the media in contact. In the polymer matrix, the processes of rearrangement of its intramolecular structure take place, while the strain hardening coefficient behaves unstable. In stage III, segments of polymer molecules acquire a certain mobility, which results in their orientation along the direction of deformation. Stage IV is characterized by the appearance of microcracks.

The shape of the AE curve reflects the division into stages and qualitative changes during transitions to the hardening stages. At small strains, dislocation glide bands and recrystallization twins are observed. In the elastic state, there are no stress concentrators capable of generating plastic shear and acoustic emission. A further increase in the load leads to powerful plastic effects, which manifest themselves in the form of slip lines. AE emission occurs between active deformation centers, which leads to the birth of a new hardening stage. Jumps of individual microcracks are detected in the form of high-amplitude AE pulses. The development of microcracks together with ongoing dislocation processes ensures the existence of the pre-fracture stage.

The mechanisms of plastic deformation can also be considered depending on the scale levels of macrorepresentations.

AE signals in the time domain demonstrate a tendency to increase in impulsivity (Figure 6 and Figure 7). Features that quantify the impulsivity characteristics of the AE signal, such as kurtosis, peak-to-peak fluctuations, crest factors, etc., are potential predictive characteristics for this data set [48].

A powerful tool for predicting the state of samples in the frequency domain is spectral kurtosis. Visualization of the spectral kurtosis of AE signals at different loads in metal-based polymer coatings is shown in Figure 10.

Therefore, statistical characteristics of AE signals in the time domain were chosen for analysis and processing, such as the mean value (Mean), standard deviation (Std), skewness (Skewness), kurtosis (Kurtosis), full swing (Peak2Peak), root mean square value (RMS), crest factor (CrestFactor), form factor (ShapeFactor), impulse factor (ImpulseFactor), marginal factor (MarginFactor), and energy (Energy), as well as statistical characteristics of the spectral kurtosis, such as mean value (SKMean), standard deviation (SKStd), asymmetry (SKSkewness), and kurtosis (SKKurtosis).

The choice of the analyzed characteristics was due to the fulfillment of the monotonicity requirement. The fulfillment of the monotonicity conditions required that the values of these characteristics exceed the level of 0.4. In addition to these characteristics, features of spectral kurtosis in the frequency domain can be calculated, such as peak and centroid frequency, which are commonly used to characterize various damage modes [49,50]. However, these characteristics do not satisfy the monotonicity criteria.

The values of these characteristics for experimental AE signals are given in digital form in Table 2.

To reduce the dimension and combine the features, the principal component analysis (PCA) was applied in the work. Principal component analysis is one of the most common factor analysis methods. It uses an orthogonal transformation of a set of observations with associated characteristics, each taking different numerical values, into a set of variables without linear correlation.

Using the Matlab 2018b computer mathematics system, when processing the data in Table 2, the first two principal components of PCA1 and PCA2 were calculated. Their mutual distribution depending on the measurement number, which correlates with the successive increase in the deflection of the sample during tests for four-point bending, is shown in Figure 11.

The numerical values of the first two principal components with a successive increase in the deflection of the sample during testing are presented in Table 3.

Figure 12 shows the changes in the values of the first and second principal components depending on the measurement number, which corresponds to a consistent increase in the deflection of the sample during bending tests.

The graph in Figure 12a shows that the first principal component (PCA1) increases almost monotonically as the deflection increases and the specimen approaches failure. Thus, the first principal component is a promising combined indicator of the state of the sample under loading. The trend of the second principal component (PCA2) in Figure 12b also shows an increase as the deflection increases but is not as clear-cut as for PCA1.

Figure 13 shows the nature of the change in the principal components of XRD1 and XDA2 depending on the deflection of the sample. A spline approximation is used for a more visual display of the change in the main components, shown on the graphs by a solid line.

The numerical characteristics of the first two principal components with a successive increase in deflection at the moments of successive fixation of AE signals revealed a linear trend of increase with increasing loading with a stable prospect of using the first principal component.

As follows from the consideration of Figure 13, the first principal component quite stably describes the general trend of loading of a polymer coating on a metal base by the AE method, while the second allows a fairly clear separation of stages II and III, which was impossible to achieve when observing the change in the AE signal during loading and dependencies *K = f*(Δ*L*).

The boundaries of stages III and IV are quite clearly traced when observing this section of the curve (Figure 13a) for the first principal component, where the character of this dependence changes.

A comparison of Figure 8 and Figure 13 reveals the identity of the hardening stages, and if in Figure 8 the division into stages was of a qualitative nature, then in Figure 13 their quantitative boundaries are indicated, obtained on the basis of experimental data processing.

Based on this, it can be noted that when the scale of structural changes in the materials under loading occurs at the macroscopic level, its only display, determined using the principal component method, can be used as a processed information characteristic of the acoustic spectrum.

## 6. Conclusions

The AE method has been quite actively used in studies of the plastic flow of metals and alloys under load; however, this method has not been previously used for continuous conjugated media in the form of metal-based polymer coatings. Performed on the basis of theoretical models and experimental measurements of four-point bending for samples coated on a metal base with synchronous measurements of the strain curve and AE, a multiscale accumulation of strains and their stages were established.The AE energy indicator of the different staging of plastic deformation of polymer coatings on a metal base is the hardening diagram constructed according to the data of a force-measuring device with a change in deflection with the synchronous implementation of AE measurements and their statistical characteristics in the space of principal components.It has been established that the linear dependence of the first principal component with a successive increase in deflection at the moments of fixation of AE signals can serve as an indicator of the state of the material of a polymer coating on a metal base under loading. The first main component makes it possible to fairly clearly separate stages III and IV of plastic deformation of metal-based polymer coatings using the AE method, while the second component allows one to quite clearly separate stages II and III.The plastic deformation of conjugated media proceeds inhomogeneously, the process involves in turn different areas of the deformable material at several structural levels, each of which is characterized by a scale determined by the nature of structural defects. The structural state of the coated material determines the nature of plastic deformation and affects the production of AE signals to the same extent. The AE of the material with coating in bending is structurally sensitive.

## Figures and Tables

**Figure 1 materials-15-03791-f001:**
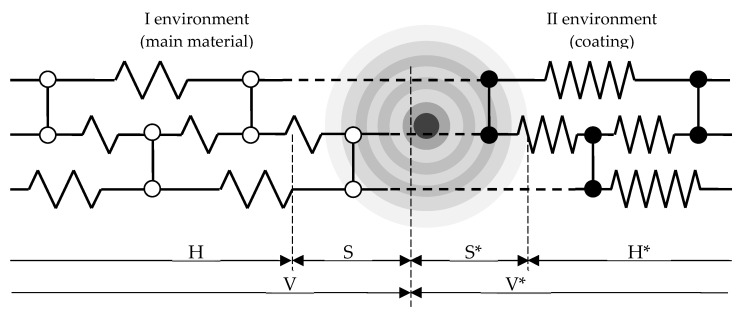
Scheme of the occurrence of AE signals in conjugated media with a point defect: ○—atoms of the 1st environment, ●—atoms of the 2nd environment.

**Figure 2 materials-15-03791-f002:**
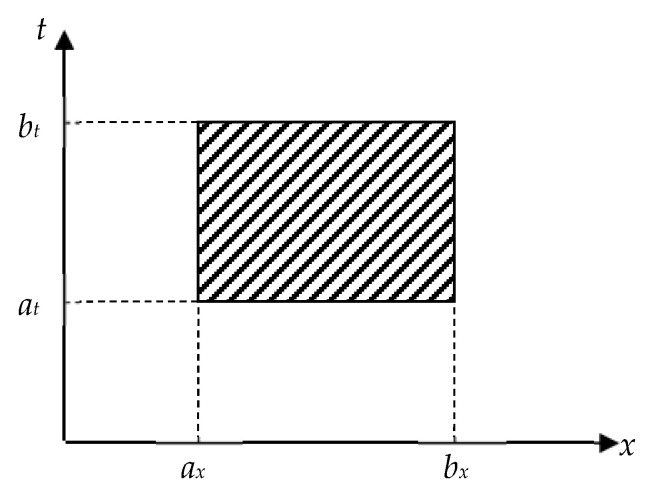
Spatial and temporal location of propagation of AE signals.

**Figure 3 materials-15-03791-f003:**
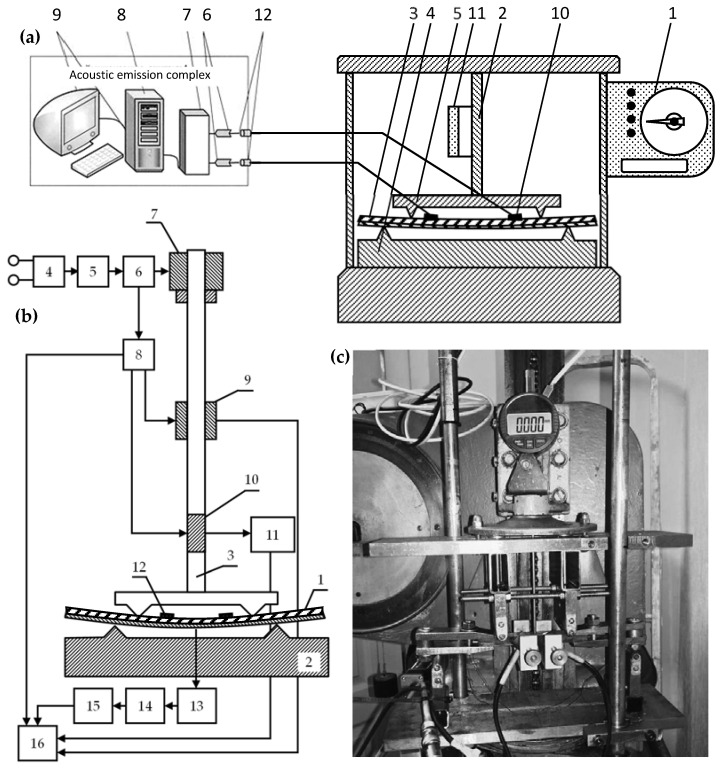
Experimental setup used for measurements: (**a**) block diagram of the setup, (**b**) functional diagram, and (**c**) general view of the setup.

**Figure 4 materials-15-03791-f004:**
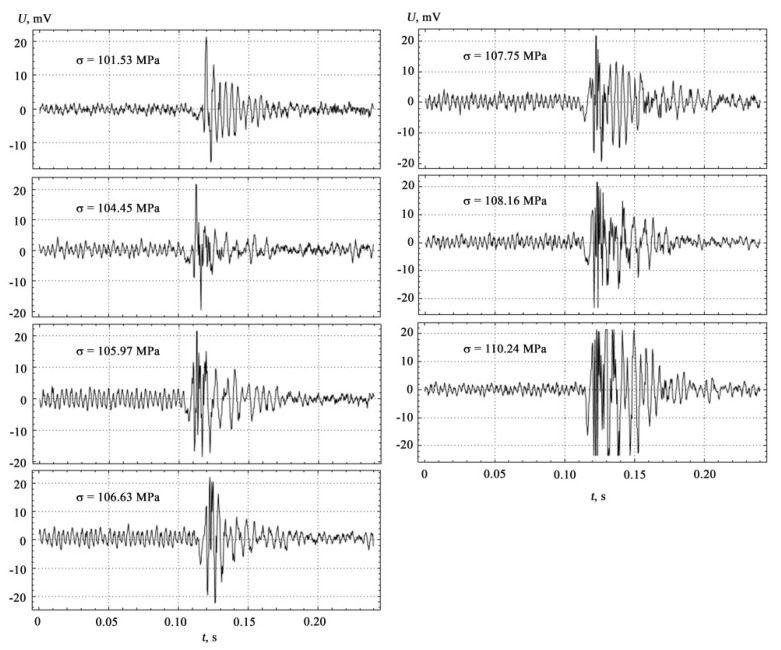
Results of AE measurements for the ED-20 polymer matrix.

**Figure 5 materials-15-03791-f005:**
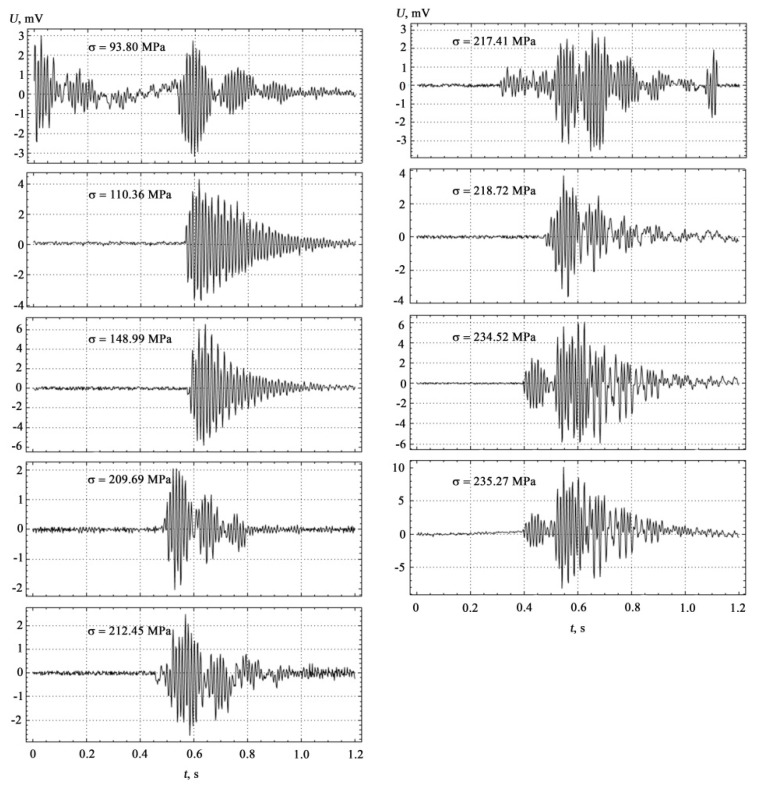
Results of AE measurements for the metal base of ST3sp steel.

**Figure 6 materials-15-03791-f006:**
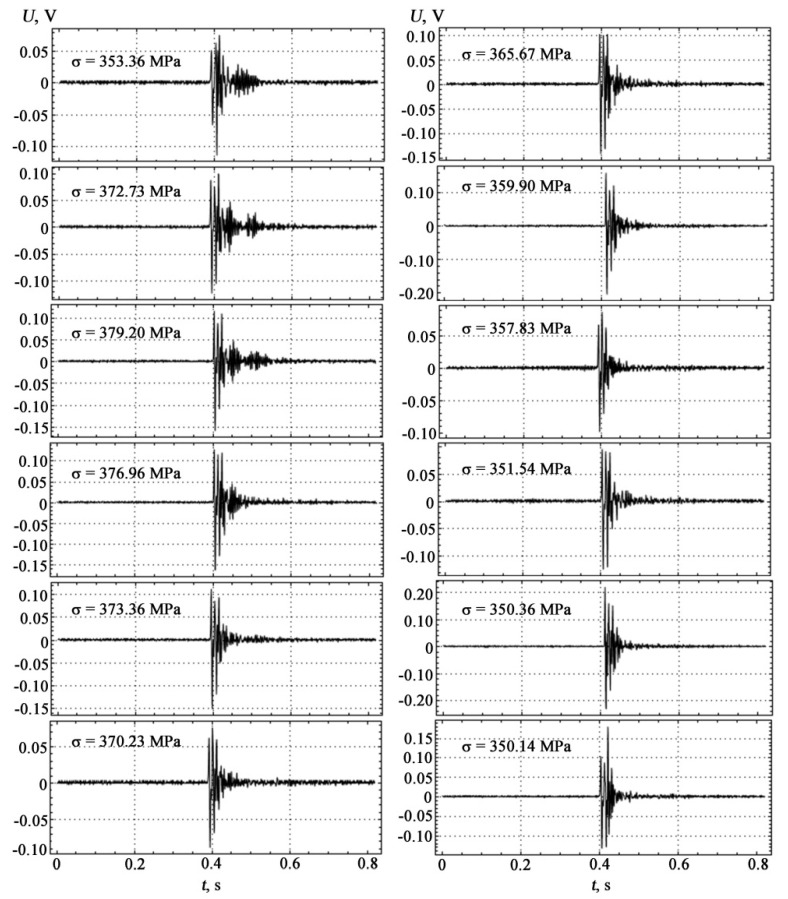
Visualization of measurement results of coated samples at different loads.

**Figure 7 materials-15-03791-f007:**
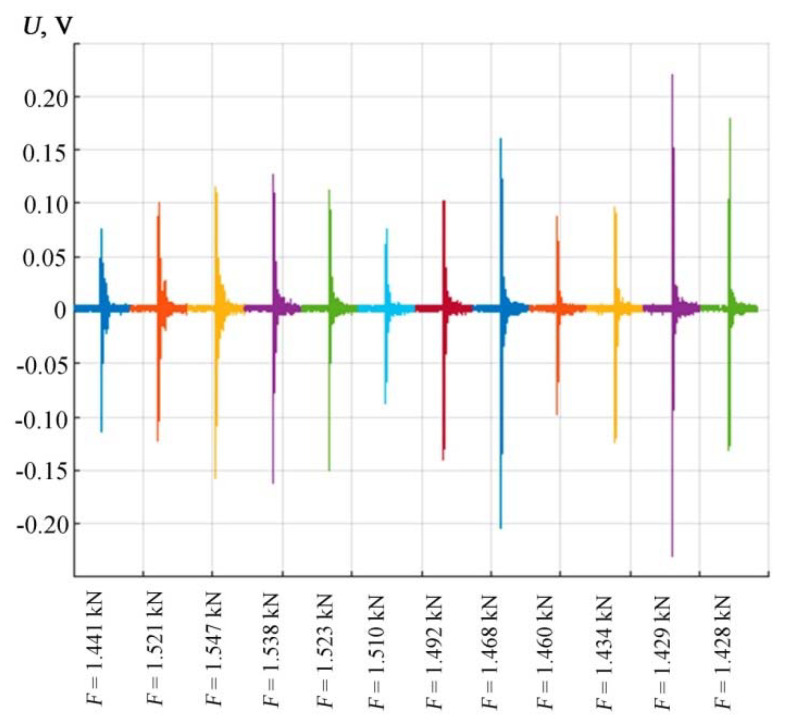
The sequence of received AE signals in the order of their occurrence and fixation during sample loading.

**Figure 8 materials-15-03791-f008:**
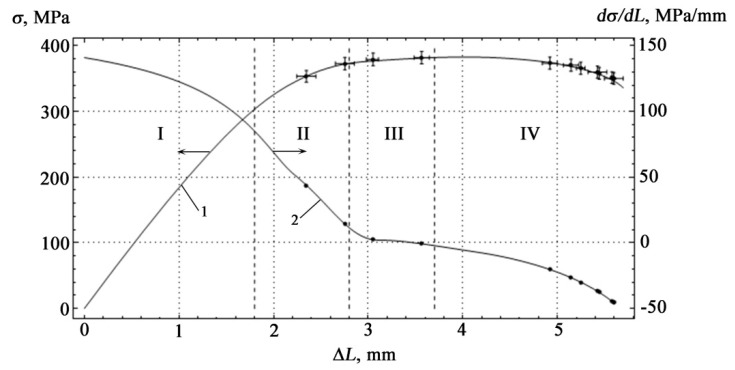
Dependence of mechanical stress (curve 1) and strain hardening coefficient (curve 2) on the deflection at four-point bending. Points on the graph show the moments of occurrence of AE signals, I–IV are the stages of strain hardening.

**Figure 9 materials-15-03791-f009:**
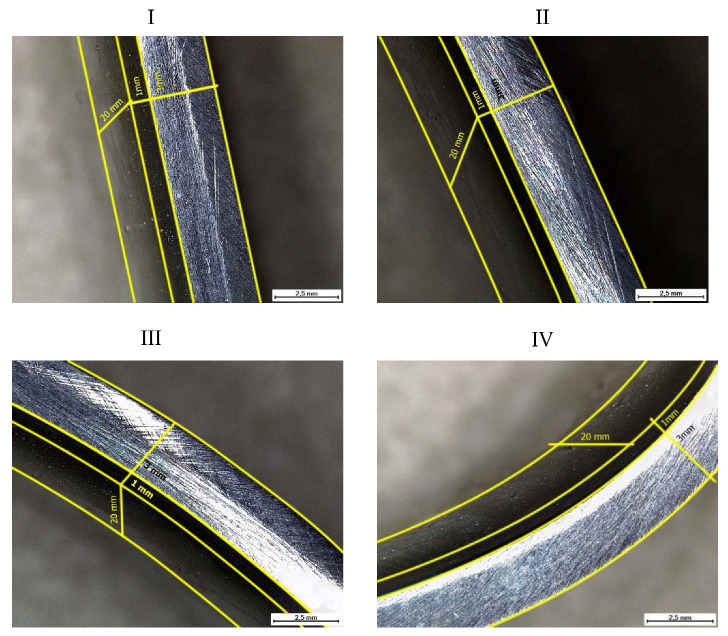
Microscopic images of the cross section of coated samples for various stages of plastic deformation under loading: I stage–*F* = 1.251 kN, Δ*L* = 1.541 mm; II stage–*F* = 1.441 kN, Δ*L* = 2.346 mm; III stage–*F* = 1.547 kN, Δ*L* = 3.052 mm; and IV stage–*F* = 1.468 kN, Δ*L* = 5.423 mm.

**Figure 10 materials-15-03791-f010:**
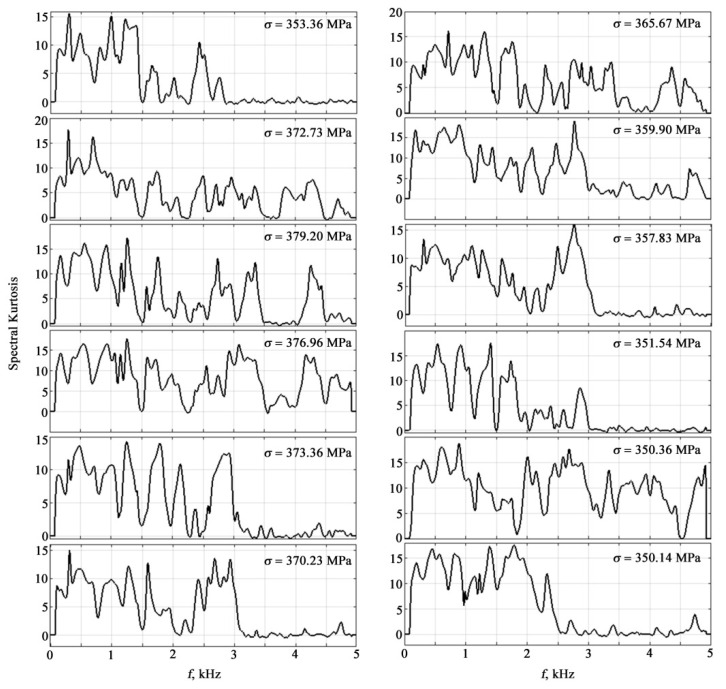
Spectral kurtosis of AE signals at different loads.

**Figure 11 materials-15-03791-f011:**
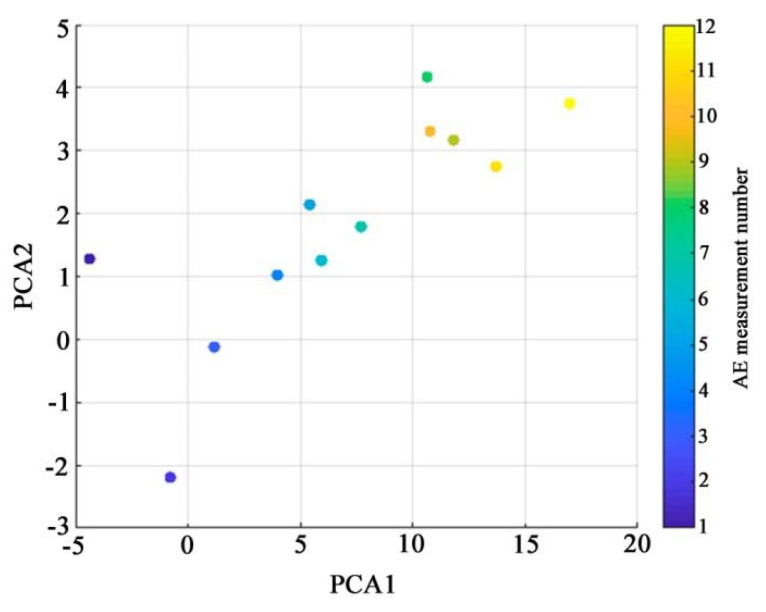
Statistical characteristics of AE signals in the space of the first two principal components (color shows the measurement number corresponding to the successive increase in the deflection of the sample during testing).

**Figure 12 materials-15-03791-f012:**
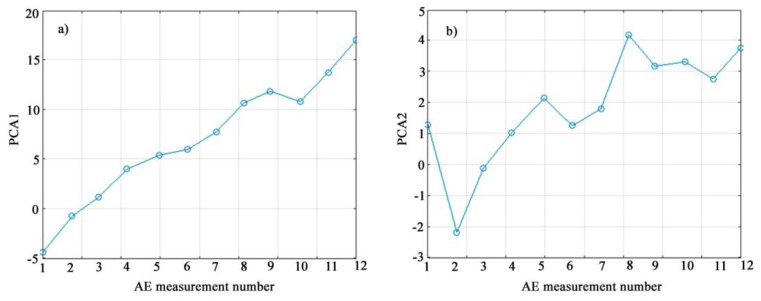
The first (**a**) and second (**b**) principal components for sequentially recorded AE signals during bending tests.

**Figure 13 materials-15-03791-f013:**
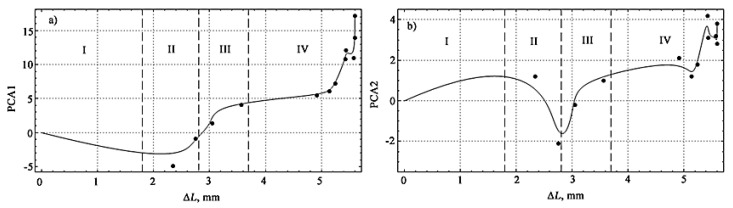
The first (**a**) and second (**b**) principal components as functions of the sample deflection (the dots show the values of the principal components for recorded AE signals, the solid line is the cubic spline approximation).

**Table 1 materials-15-03791-t001:** Experimental values of the main characteristics during four-point bending of samples at the moments of successive fixation of AE signals.

№ AE Measurements	Loading Force *F*, kN	Mechanical Stress *σ*, MPa	Sample Deflection Δ*L*, mm	AE Signal Amplitude *U*_max_, V
1	1.441	353.367	2.346	0.114
2	1.521	372.737	2.753	0.122
3	1.547	379.201	3.052	0.158
4	1.538	376.967	3.565	0.162
5	1.523	373.362	4.920	0.150
6	1.510	370.238	5.144	0.088
7	1.492	365.671	5.246	0.140
8	1.468	359.904	5.423	0.204
9	1.460	357.837	5.442	0.098
10	1.434	351.541	5.578	0.124
11	1.429	350.363	5.595	0.230
12	1.428	350.147	5.596	0.180

**Table 2 materials-15-03791-t002:** Statistical characteristics of AE signals in the time and frequency domains.

Characteristic	No. of AE Signal											
	1	2	3	4	5	6	7	8	9	10	11	12
Mean	0.0015	0.0014	0.0014	0.0014	0.0014	0.0013	0.0013	0.0013	0.0013	0.0013	0.0013	0.0013
Std	0.0093	0.0103	0.0112	0.0120	0.0119	0.0112	0.0117	0.0123	0.0116	0.0111	0.0123	0.0132
Skewness	−1.578	−1.228	−1.213	−1.253	−1.358	−1.170	−1.195	−1.382	−1.188	−1.230	−1.125	−0.957
Kurtosis	46.149	45.482	47.663	50.458	53.994	53.557	55.731	61.046	61.874	62.314	63.374	65.952
Peak2Peak	0.1900	0.2060	0.2287	0.2440	0.2476	0.2337	0.2423	0.2660	0.2513	0.2397	0.2710	0.2957
RMS	0.0094	0.0104	0.0113	0.0120	0.0119	0.0113	0.0118	0.0124	0.0117	0.0112	0.0123	0.0133
CrestFactor	8.0987	8.4075	8.5545	8.6761	8.8703	8.9474	8.9696	9.3027	9.5054	9.4339	9.7895	10.389
ShapeFactor	2.4808	2.6267	2.6870	2.8167	2.8704	2.8598	2.9581	3.0764	3.0908	3.0715	3.1648	3.2733
ImpulseFactor	20.092	22.129	23.034	24.521	25.570	25.674	26.538	28.729	29.444	29.048	31.244	34.330
MarginFactor	0.0053	0.0056	0.0055	0.0058	0.0062	0.0067	0.0068	0.0073	0.0080	0.0081	0.0082	0.0086
Energy	0.1442	0.1799	0.2138	0.2424	0.2381	0.2165	0.2344	0.2593	0.2333	0.2125	0.2709	0.3057
SKMean	3.4311	4.1231	4.6108	5.4668	5.3253	5.1467	5.6013	5.9133	5.7547	5.1543	5.9802	6.2471
SKStd	4.5471	4.0932	4.3303	4.4467	4.5093	4.4905	4.4515	4.7328	4.6984	4.7966	4.7387	5.0351
SKSkewness	1.0117	0.7975	0.7057	0.5134	0.5035	0.4947	0.3693	03441	0.3282	0.4864	0.3356	0.3364
SKKurtosis	2.6877	2.8859	2.6144	2.4438	2.2819	2.1841	2.0602	1.8779	1.8289	1.8932	2.0574	2.0342

**Table 3 materials-15-03791-t003:** Values of the first two principal components of PCA1 and PCA2 for four-point bending of specimens at the instants of successive fixation of AE signals.

№ AE Measurements	Mechanical Stress *σ*, MPa	Sample Deflection Δ*L*, mm	PCA1	PCA2
1	353.367	2.346	−4.912	1.230
2	372.737	2.753	−0.925	−2.152
3	379.201	3.052	1.412	−0.215
4	376.967	3.565	4.157	1.011
5	373.362	4.920	5.540	2.120
6	370.238	5.144	6.172	1.213
7	365.671	5.246	7.256	1.812
8	359.904	5.423	10.847	4.250
9	357.837	5.442	12.129	3.190
10	351.541	5.578	11.012	3.205
11	350.363	5.595	13.904	2.819
12	350.147	5.596	17.218	3.876

## Data Availability

The data presented in this study are available on request from the corresponding author.

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
