# Peer review of "Experimental and Theoretical Study of Plastic Deformation of Epoxy Coatings on Metal Substrates Using the Acoustic Emission Method"

_materials, 2022, doi:10.3390/ma15113791_

Round 1

Reviewer 1 Report

The paper was written in a very thoughtful manner. The research topic has been presented thoroughly and the content of the paper is not objectionable. After reviewing the content of the work and the enormous contribution of the authors to its creation, I conclude that the work can easily be published in its current form, due to the quality and manner of the content and research results presented in it.

Author Response

Thank you for evaluating and reviewing the manuscript, which, after improvements, is presented in the attached file.

Reviewer 2 Report

The positive thing in this work is the experimental setup, which could be interesting to be explored. The insitu experimental setup which correlates the acoustic emission measurements with the deformation of a coated metal substrate/base is relevant. At moment, the work is not suitable for publication. There are serious problems with the structure of the manuscript. The experimental part needs much more clarity and description. The overall cohesion requires some improvements. The work needs at very least a significant major revision. English level has to be improved as well.

Title:

  • What I understood is that you are using Epoxy coatings on top of steel substrates. A metal-based polymer coating is another thing.

Introduction:

  • it is not clear what is the scientific question/problem the authors are addressing. There is much text about the befits of AE in different applications, but any description of the scientific gap. You did not state why your work is relevant or what is your main contribution to the next step of acoustic emission (AE) measurements/modelling?

  • The aim of the work is a bit confusing. Your title is about the deformation of polymer coatings using AE, but you stated in line 79 that AE is being performed during the deformation of metal samples?

  • The authors have to describe how they tackled the scientific problem at the end of the introduction. It is not there.

Literature review:

  • In the journal, the literature review is normally included in the introduction, and not a separated section. There are many problems here as well. Apart from the poor English level, the authors are just listing several works on EA without any connection with the present work. The idea of a literature review in a scientific paper is to tell the readers the main development until your contribution, and not list all papers on AE you found. I suggest the authors rewrite it by selecting suitable references and incorporating the text in the introduction. Please, check out recent publications in the journal too.

  • The last part of the literature review (lines 161 to 164) seems that it has to be placed at the end of the introduction. Is that the problem you want to bring up?

Material and method:

  • Very poor description of the sample preparation and the material constituents e.g. brands of the materials, how the samples were prepared, if possible, microscopy images of the cross-section where the readers can see the coating-interface-steel structure. All of these aspects must be reported, otherwise, the work is not reproducible.

  • In lines 191-193, the authors mentioned the method of principal components. Please, clarify what that is in detail because it was one of the key things for AE measurements?

  • In table 1, some units are in MPa and others in N/mm^2. Please, be consistent and choose only one since they are equivalent.

  • There are some ISO standards throughout the paper, but they are not in the reference list.

Experiments:

  • The authors mentioned in lines 198-199 that the tests are to measure plastic deformation on the coating on metal base/substrate, but the work itself seems that you are measuring the bending stress in the coated steel. Please clarify in the manuscript whether you are focusing on the coated samples or the coating only?

  • How many samples did you test? How about the error e.g. standard deviation?

  • There is any information about the thickness of the coating. This is crucial information when coatings affect the deformation of the coated substrate. This must be stated and measured.

  • Fig. 3 does not show the stress transfer mechanism between the coating and metal substrate/base. Is that the classical shear lag model? Why did you not explain that?

  • Eq. (1) is from the beam theory. Please cite it properly. You did not introduce it in this work!

  • Fig. 4 is just a photo of the experimental setup. My suggestion is to explain the components. Make some arrows on that. Even better to prepare a schematic image of the experimental setup. From line 235, you discussed how the position of the AE sensor affects the measurements. You must show it more clearly to the readers. Schematic image?

  • There are no images of the samples before and after tests to show the damage mechanisms e.g. cracking on the coating, interfacial delamination, microcracks and etc. When you explain the stages in Fig. 10 (from line 427) is all about speculation. How can you model something without showing the mechanism? Is that possible also to use some references to support your arguments?

  • In Fig. 5, t stands for time, but you also have t in Fig. 3 standing for the thickness of the beam. Please be consistent all over your work!

Results and discussion:

  • The authors have written the whole model in this section. It is not the right place for it, is it?

  • I guess some of the equations are from literature as the equations of motion. You must cite them.

  • In the equations of motion, how the coordinate system is related to your work? And why are you stating all equations if you are basically using only one component (bending stress)? If you try to make it shorter by stating the necessary equations, it would slightly improve the understanding of the model. At moment, the presented way is not clear whatsoever.

Conclusion:

  • I guess what you are trying to do is to correlate damages in the coating with the plasticity in the metal substrate/base. This is also related to the thickness of the coating. The title indicates that your method is to measure the behaviour of the polymer coatings. You must restructure this work, and rethink how to organize it overall.

Author Response

According to the wishes of the reviewer

- The title of the article has been changed. So it more adequately reflects the essence of the work

- The purpose and scientific direction of the work is more clearly formulated, which consists in a deeper use of the principal component method in assessing the staging of deformations of multilayer structures and the boundaries of their changes.

- The Introduction and Literature Review sections are merged with the addition of the latest publications in the journal on the use of the principal component method in assessing the staging of deformation based on the results of AE measurements. Corrections and corrections have been made to the text of the literature review. The problem to be solved is formulated at the end of this section.

- The sample preparation for testing is described in detail, indicating the modes of coating application. An explanation is given on the use of the principal component method, which is used as the main method for analyzing the results of AE measurements. Its advantages and methods of application are indicated.

- Table 1 has been corrected in accordance with the wishes of 4 reviewers

- Made references to ISO standards

- In the Materials and Methods section, a theoretical model was added with the addition of information from the theory of beams. Appropriate references are made in support of the arguments of the staging of deformations and the spectral analysis of AE signals under loading of multilayer structures. This is done to preserve the logic of the presentation.

- The description of the experimental setup has been strengthened, indicating the structural positions and their functioning during the measurement process. Two drawings have been added. The text of this section includes information about the conditions of the experiment.

- The Results and discussion section begins immediately after the description of the experimental setup. Now it includes fig.5,6 and table.4. Added material on the stages of plastic deformation and the corresponding mechanisms for their description with references to literary sources.

In general, the work has been restructured, rethought and organized according to the new edition.

Reviewer 3 Report

Overall, this manuscript is not articulate and the writing needs extensive improvements. The grammatical errors made the sentences hard to read. The contents are scattered instead of focusing on the novelty and validation of the testing method. The logic flow is disturbed through the article. Given these shortcomings the manuscript requires major revisions. Specific comments follow.

Abstract

Line 13: The sentence should be written in passive instead of active voice. Please change the sentence to “The propagation of xxx was explored”.

Line 14: Same with the issue mentioned above.

Line 18: The same issue as mentioned above.

Line 20: A “period” should be added between “structures” and “Boundary”.

Line 27: Please change to “In this work, we presented”.

Keywords

The keywords are not representative to highlight the theme.

Introduction and Literature review

It is unnecessary to separate the introduction and literature review sections. It will be easier for the readers to understand the importance of this work with the context of the progress in this field.

Figure 1 and 2 can be deleted or put in the supplementary information.

Table 1, 2, and 3 can be put in the supplementary information.

Figure 5, 6 and Table 4 should be put in the “Results and Discussion” section. 

Figure 6, 7 should be put in the supplementary information.

The data in Table 5 should be presented within a figure.

Author Response

- The title of the article has been revised, taking into account the specification of the materials.

- The goal, the provisions of novelty are clearly formulated, the scientific direction of research is highlighted, which consists not only in the use of new samples of measuring equipment, but also in the use of new forms of processing information parameters that are sensitive to changes in multilayer structures under loading, through the use of the principal component method.

- Restructured Abstract section

- Sections Introduction and Literature Review merged. The literature review is supplemented with appropriate links indicating the progress in using the principal component method in processing AE signals and extracting additional information about the staging of plastic deformation processes.

- Fig.1,2 removed from the text and replaced by a corresponding reference to our work previously published in this journal [35]

- Tables 1,2,3 containing reference data on the properties of the substrate and coating material, instead of a detailed illustration, are replaced by the corresponding links [1,2]

- A detailed analysis of Table 6 is given with the rationale for the need for its use and the requirements for the monotonicity of the incoming components

- The content and style of presentation have been radically revised and improved:

Theoretical models have been moved to the Materials and Methods section

To preserve the logic of the flow and the general focus of the presentation on the study of plastic deformation of epoxy coatings on metal substrates using the acoustic emission method and signal processing using the principal component method, it is shown that the use of the principal component method is a necessary stage for obtaining input information for the mathematical processing of AE signals in the system diagnostics of destruction and damage of materials under load. In this case, new possibilities appear for separating weak signals even in the case when their amplitude is significantly lower than the noise level.

Theoretical models are made with the addition of information from the theory of beams. To preserve the consistency of the presentation, they have been moved to the Materials and Methods section.

The Results and Discussion section begins immediately after the description of the experimental setup and now includes Figs. 5, 6 and Table 4.

In general, the work has been restructured, rethought and organized according to the new edition.

Reviewer 4 Report

This paper investigated the characteristics of AE signals generated during the deformation of metal-based polymer coatings for identifying the damage state of the material. The topic of this paper is interesting. However, before its publication, some major revisions are needed.

  1. Show the positions of AE sensors and polymer coating deposited on the metal base in test schema in Fig. 3.
  2. How did you measure the crack initiation and propagation of the coating? Did the cracks occur in the polymer coating during your bending test?
  3. The method of eliminating noises should be provided in the Experimental section.
  4. Why the authors only show 16 signals recorded at different loads (Fig. 5) instead of all signals recorded during bending test? How many
  5. The authors performed the statistical analyses of AE features in both time and frequency domains, which is very important. However, it must be admitted that the data set was too small since there were only 16 signals.
  6. In Line 452, the authors calculated the feature (spectral kurtosis) in the frequency domain. However, two frequency domain parameters including the peak frequency and centroid frequency were neglected, which are commonly used for characterizing different damage mode. The authors can refer to the following links.

https://doi.org/10.1016/j.ijfatigue.2022.106860

https://doi.org/10.1016/j.jmapro.2022.01.012

  1. Four stages were discriminated based on the changes of principal components (Fig. 14). What did each stage exactly mean? Moreover, no AE signal was observed in stage 1. Was it reasonable to divide the deformation process into four stages?

Author Response

- The position of the AE sensors and the polymer coating on the metal base in the test scheme is reflected in two new figures5,6 of the new edition.

- In the experiment, the coating thickness was fixed. There is a relationship between the plasticity of the metal substrate and damage to the coating under loading, due to the thickness of the coating. The consequence of this is cracking on the coating, interfacial delamination, microcracks, etc. This is an important scientific area of research in the field of materials science, structural mechanics and the theory of strength of conjugated media. Its consideration goes beyond the scope of more than one similar scientific article and will undoubtedly be the basis for further research.

- During the experiment, 657 AE signals were registered. Figure 10 shows signals that exceed the minimum threshold level of 0.01 V.

- The choice of the analyzed characteristics was due to the fulfillment of the monotonicity requirement. The fulfillment of the monotonicity conditions required that the values of these characteristics exceed the level of 0.4. In addition to these characteristics, features of spectral kurtosis in the frequency domain can be calculated, such as peak and centroid frequency, which are commonly used to characterize various damage modes [49,50]. However, these characteristics do not satisfy the monotonicity criteria.

- The hierarchy of structural levels of plastic deformation of solids can vary from three stages, including elastic deformation, elastic-plastic and destruction, up to five or more, depending on the structure of the material.

Round 2

Reviewer 2 Report

Still, some issues with the structure of the manuscript are found. There is no subsection for the model. Their literature review is just a list without supporting the relevance of the current work. For example, in lines 84 onwards:

In [3,4], the results of experimental studies on the identification of the propagation of acoustic emission oscillations in an epoxy polymer matrix are presented.

It needs some explanation not just presenting somebody’s work. Fig. 4 there are texts that are not in English.

  • In lines, 398-399 is about the mechanism of the metals, not the coating you are studying. It is still not clear if AE measurements are really from the coating or metals substrate.

  • Fig. 2 is just from textbooks and the stress is from the metal, not from the coating. The work is related to the coating.

  • How many samples did you test? How about the error e.g. standard deviation? There is a need to show microscopy images of the cross-section where the readers can see the coating-interface-steel structure, and images of the damage in the coating of each stage to be able to make any correlation. All of these aspects must be reported, otherwise, the work is not reproducible.

The experimental part must improve more, otherwise, the correlation between the model and one experiment is more or less speculation.

Author Response

In accordance with the comments of the reviewer

- the structure of the literature review has been revised and rethought with an emphasis not only on stating the sources of the background of the work, but also on their connection with certain aspects of the proposed manuscript and their concretization for use in this work

- drawing with a general scheme of a four-point bend is removed as well-known

- inaccuracies associated with the presentation of information not in English were eliminated in the description of the block diagram of the experimental setup

- separately presented and considered in detail the subsection of modeling the process of changes in the structure of multilayer conjugated media during bending with the study of concomitant changes in the spectrum of AE signals

- in the Modeling subsection, the characteristic features of the AE spectrum under loading are highlighted, which are the basis for the correlation of the model with the experiment

- the experimental part has been improved, the results of complex mechanical tests of the flexural loading diagram and measurements of the acoustic spectrum, performed separately on the ED-20 epoxy matrix used as a coating, and on a metal substrate made of ST3sp steel are described in detail and given.

- a comparative assessment of the moments of occurrence of AE signals in a four-point bending on a coating, a metal substrate and a joint conjugate medium was performed

- in the course of the experiments, five coated samples were tested, the errors in determining mechanical stress and deflection were calculated and given

- microscopic images of the cross section of coated samples in different zones of plastic deformation under loading are given

Reviewer 3 Report

Thanks to the authors' work, the writing has been improved and the layout is cleaner. Changing the title makes the topic of this manuscript much more clear. It is a good way of using the principal component analysis for the acoustic-emission signals to investigate the status of organic coatings on metal substrates. However, the writing is still rough and there is room for improvement. Specific comments follow.

Introduction

Line 37-73, the first five paragraphs of the Introduction section was not concise. Reducing the length will make the logic flow look better.

At the beginning of the Introduction section, I believe the authors were trying to express this:

“The polymer coatings are necessary to protect the metals from corrosion and oxidation. It is important to monitor the status of the polymer coatings in the long-term applications. With xxx properties, epoxy polymers are among the best coating candidates. So the epoxy coatings were studied in this research. The acoustic emission method is efficient to detect the failure of the polymer coatings.” And then, the references can be introduced to review the work done in this field, and gradually bring out why this research is novel among them.

Line 44, “Improved technological properties characterize epoxy oligomers” is not a complete sentence. Deleting this sentence will make the writing flow better.

Materials and methods

Line 157-173 can be put in the Introduction section.

Only the elaboration of the methods being used in this manuscript is necessary in the Methods section.

Figure 2, 3 and 4 can be combined into one large figure to make the presentation of the experiments more clear.

Results and discussion

Line 568-569, “the scale of functional changes in the structure” was not expressed clearly. It’s not the functional changes, but the structural changes of the polymer chains were caused by the accumulative stress in the environment.

Conclusions

Line 580, should be changed to “indicators of the different stages of ”.

Line 595-597, does the coated material mean the coatings? If yes, then should be changed to coating material.

Author Response

In accordance with the comments of the reviewer

- the logical scheme of the “Introduction” section has been improved, the introductory part has been shortened, the relevance of the work has been more clearly indicated, the need for research and the novelty of approaches to its solution have been outlined

- the incomplete line 44 is excluded from the text, which makes the presentation of the material more correct

- the wording of the last paragraph before the “Conclusions” section was changed and clarified, “scales of functional changes” were replaced by “scales of structural changes”. From the point of view of accumulative stresses during bending, the reviewer's proposal is fair and valid.

- for greater clarity, the representations of the experiment are combined into one general drawing of the experimental setup, a block diagram of the setup, a functional diagram and its general view

- conclusion 2 clarifies the terminology of the staging part with an emphasis on indicators of different stages of plastic deformation

Reviewer 4 Report

After revision, this manuscript can be accepted in present form.

Author Response

(The authors gave the same response as above.)
